# Navigating the Chemical Space of ENR Inhibitors: A Comprehensive Analysis

**DOI:** 10.3390/antibiotics13030252

**Published:** 2024-03-11

**Authors:** Vid Kuralt, Rok Frlan

**Affiliations:** Department of Pharmaceutical Chemistry, Faculty of Pharmacy, University of Ljubljana, 1000 Ljubljana, Slovenia; kuraltvid@gmail.com

**Keywords:** antibacterial drug development, enoyl-acyl carrier protein reductases, ENR inhibitors, antibiotic resistance, Fab, InhA, bioinformatical analysis

## Abstract

Antimicrobial resistance is a global health threat that requires innovative strategies against drug-resistant bacteria. Our study focuses on enoyl-acyl carrier protein reductases (ENRs), in particular FabI, FabK, FabV, and InhA, as potential antimicrobial agents. Despite their promising potential, the lack of clinical approvals for inhibitors such as triclosan and isoniazid underscores the challenges in achieving preclinical success. In our study, we curated and analyzed a dataset of 1412 small molecules recognized as ENR inhibitors, investigating different structural variants. Using advanced cheminformatic tools, we mapped the physicochemical landscape and identified specific structural features as key determinants of bioactivity. Furthermore, we investigated whether the compounds conform to Lipinski rules, PAINS, and Brenk filters, which are crucial for the advancement of compounds in development pipelines. Furthermore, we investigated structural diversity using four different representations: Chemotype diversity, molecular similarity, t-SNE visualization, molecular complexity, and cluster analysis. By using advanced bioinformatics tools such as matched molecular pairs (MMP) analysis, machine learning, and SHAP analysis, we were able to improve our understanding of the activity cliques and the precise effects of the functional groups. In summary, this chemoinformatic investigation has unraveled the FAB inhibitors and provided insights into rational antimicrobial design, seamlessly integrating computation into the discovery of new antimicrobial agents.

## 1. Introduction

When antibiotics began to be used to treat bacterial infections, it quickly became clear that the bacteria would fight back, developing resistance to the antibiotics currently in use (so-called drug-resistant (DR) and multidrug-resistant (MDR) bacteria). The treatment of drug-resistant infections is very expensive and time-consuming, and can have serious or even life-threatening side effects. For example, the average direct cost in the U.S. for 2020 is between $20,000 for the treatment of drug-resistant TB and $568,000 for the treatment of extensively drug-resistant tuberculosis (XDR TB) [1]. This requires the research, discovery, and development of new antimicrobial active pharmaceutical ingredients (APIs). Enoyl-acyl carrier protein reductases (ENRs) are one of the most promising targets for the treatment of bacterial infections. The most abundant enzyme is FabI, while other isoforms such as FabK (e.g., the only enoyl reductase enzyme in *Streptococcus pneumoniae*), FabL (*Bacillus subtilis* has FabL and FabI), FabV, and InhA (also known as mycobacterial FabI (MtFabI)) are less common. ENRs are important enzymes in the bacterial biosynthesis of fatty acids (FAB-II pathway) for cell wall formation and are distinct from those in mammals (so-called FAB-I pathway), which is important for selective toxicity [2]. Despite some debate about their essentiality [3], ENRs remain important targets for the development of antistaphylococcal [4], antimalarial [5], and antibacterial agents, particularly against Gram-negative bacteria and against *M. tuberculosis* [6].

Most ENRs belong to the superfamily of short-chain alcohol dehydrogenases/reductases (SDRs) and catalyze the final and rate-limiting step of fatty acid elongation, namely the reduction of the C–C double bond of the enoyl intermediate. They reduce the double bond in trans-2-enoyl-ACP to acyl-ACP with the help of the co-factor NADH, which leads to hydrolysis to the final product—a fatty acid (Figure 1) [2]. Although they catalyze the same reaction, the active sites of the ENRs, particularly FabK and FabV, differ, with InhA and FabI showing the greatest homology. Both InhA and FabI exhibit homotetrameric structures with a conserved SDR motif (Tyr-X6-Lys). In this motif, lysine interacts with the co-factor NADH via a hydrogen bond to ensure the stability of the complex, while tyrosine acts as a proton donor (Figure 2). The crystal structures of *E. coli* FabI and *M. tuberculosis* InhA show that InhA has a deeper binding cleft than FabI, which allows it to accommodate long-chain substrates that bind in its long hydrophobic tunnel during mycolic acid biosynthesis. *B. subtilis* FabL and *E. coli* FabI have only 25% sequence similarity. Furthermore, while *B. subtilis* FabL utilizes NADPH as a cofactor, *B. subtilis* and *E. coli* FabI utilize NADH. Although FabL has the same Tyr-X6-Lys motif as FabI, it has a distinct coenzyme binding site [2,7,8].

FabV of *V. cholerae* is 60% larger than other typical SDR enzymes, which are usually about 250 residues long. In addition, the active site spacing of FabV consists of eight residues (Tyr-X8-Lys), in contrast to FabI and FabL, which have a spacing of six residues. However, FabV contains the same co-enzyme binding site as FabI, which uses the classical Rossman fold motif. FabK from *S. pneumoniae* is an FMN-dependent oxidoreductase that belongs to the family of NAD(P)H-dependent flavin oxidoreductases. NADH serves as a reducing agent, but acts indirectly by reducing the tightly bound flavin cofactor. This reduced flavin then reduces the double bond [7].

There are two widely used inhibitors of ENRs today, a trichlorinated biphenyl ether called triclosan, which inhibits FabI of various bacteria as well as *M. tuberculosis* InhA, and isoniazid (INH), which also inhibits InhA (Figure 3). Triclosan is a synthetic compound that is very commonly used in everyday products such as soap, toothpaste, and plastics, while INH is widely used as an anti-tuberculosis drug. Although both agents have been used for decades, ENRs tended to be neglected until the late 1990s when it was recognized that InhA and FabI were primary targets of isoniazid and triclosan, respectively [9]. This led to an increased interest in these enzymes in the last two decades, which was favored by the resistance to both drugs and the poor pharmacokinetic properties of triclosan. In the past decades, other modified derivatives of INH and especially of triclosan with improved pharmacokinetics have been synthesized, as well as many new drugs with completely new structural properties. These have been described in patent applications [6] and extensive literature reviews [2,10,11] in the last 10 years.

Our preliminary analysis of interest in the individual enzymes (Figure 4) shows minimal research interest in inhibitors for FabV (green color) and FabK (orange color). However, there is notable interest in researching inhibitors for FabI (blue color) and InhA (red color), with a recent focus on InhA. The most productive years for Fab enzyme research were 2006, 2007, and 2014 in terms of the total number of compounds published in each year between 2001 and 2022.

Although some of these potent inhibitors have progressed to Phase II clinical trials, none of them have yet received clinical approval. Most of these inhibitors have been identified by high throughput screening (HTS) and to a lesser extent by screening of natural products or virtual screening [6]. Due to resistance to existing treatment options, there is an urgent need for new chemical agents that target bacteria, and ENRs are valuable targets; therefore, there is a clear need for new approaches to address this problem.

To close this gap, it is essential to integrate various tools from the fields of bioinformatics, chemoinformatics, machine learning, and chemometrics. The use of existing data on known inhibitors and their systematic analysis through techniques such as quantitative structure–activity relationships (QSAR) and various machine learning approaches are crucial. At the heart of this methodology is chemometrics, which applies mathematics, statistics, and formal logic to explore the chemical space and drive scientific discovery. It encompasses various techniques, including multivariate models such as principal component analysis (PCA), partial least squares (PLS), and bilinear techniques such as principal component regression (PCR) and partial least squares discriminant analysis (PLS-DA). These interdisciplinary methods can help researchers unravel complex relationships within chemical datasets and identify key physicochemical properties and molecular features relevant to a compound’s bioactivity [12,13,14,15,16]. In addition, these techniques can predict unexplored properties not covered by HTS or virtual screening and thus have the potential to complement more traditional methods and revolutionize drug discovery. The knowledge gained can help in the development of novel chemical entities with new scaffolds and properties, thus contributing to the advancement of ENR inhibitors and addressing the urgent need for effective treatments [17,18].

Therefore, our current study is dedicated to analyzing the predominant chemical space of ENR inhibitors. To this end, we have compiled a comprehensive dataset comprising 1412 small molecules recognized as ENR enzyme inhibitors. These were obtained from ChEMBL, BindingDB, our internal database from previous projects, and available literature data (Appendix A). Our primary goal was to rapidly yet thoroughly investigate the chemical landscape of existing ENR inhibitors using advanced cheminformatics tools. Through this investigation, we aimed to explore the physicochemical properties of known ENR inhibitors and identify features that are important for activity, thereby improving our understanding of the molecular structure and features that influence the structure–activity relationship (SAR) being sought. Our overarching goal is that this knowledge will aid in the identification of new potential drug candidates against bacterial infections, ultimately reducing failure rates in the later stages of drug design and development.

## 2. Results and Discussion

### 2.1. Data Collection, Preprocessing, and Classification

After a careful curation process described in the Materials and Methods section, the final dataset comprised 1412 small molecules (Appendix A) with pIC_50_ values ranging from 3.5 to 9.7. To distinguish between active and inactive compounds, we used a threshold of 5.5 for the pIC_50_ value. This decision was made in view of the fact that many articles in this field have chosen threshold values between 5 and 6 [17,19,20]. Furthermore, the choice of 5.5 as a threshold resulted in a distribution of 855 (61%) inactive and 557 (39%) active compounds in our database, ensuring a balanced representation of both categories. Compounds below this threshold were categorized as inactive, while those that exceeded it were considered active (Appendix A).

### 2.2. Physicochemical Properties

In our analysis, we looked at the physicochemical properties of the inhibitors in each enzyme dataset and presented the key results in Figure 5. A total of 39 features were generated using RDKit [21], including enhanced/hybrid logP (SlogP), number of aromatic rings (NumAromaticRings), number of stereocenters (NumStereocenters), total polar surface area (TPSA), average molar weight (AMW), number of rotatable bonds (NRB), number of hydrogen bond donors (HBD), number of hydrogen bond acceptors (HBA), fraction of carbon atoms that are sp3 hybridized (FractionCSP3), and many others. For a comprehensive understanding, the results of the statistical analysis are presented in Appendix A.

Our comparisons revealed remarkable differences between the InhA and FabI inhibitors, with distinct features emerging. InhA inhibitors exhibited significant features, such as higher SlogP, a greater NumAromaticRings, NumStereocenters, and FractionCSP3. In addition, they exhibited a higher HBD and NRB compared to FabI inhibitors, combined with a higher sum of atomic polarizabilities (apol) and a smaller radius. Understanding these differences in physicochemical properties is critical. These differences arise from the unique structural features of InhA, particularly its deeper binding cleft and the need to accommodate long-chain substrates during the biosynthesis of mycolic acid [7]. Consequently, InhA inhibitors may require higher lipophilicity to interact effectively with the hydrophobic regions within the binding site, and the smaller radius observed may be attributed to the specific geometric constraints imposed by the deeper binding cleft. In addition, the presence of a long hydrophobic tunnel in InhA requires that its inhibitors have a higher number of aromatic rings, stereocenters, and rotatable bonds. These features contribute to the flexibility and adaptability of the molecules within the binding site.

On the other hand, FabK inhibitors showed a different profile compared to FabI and InhA inhibitors, with higher values for NumAromaticRings, HBD (hydrogen bond donors), NRB (number of rotatable Bonds), amide bonds, the number of aliphatic rings, TPSA (topological polar surface area), SlogP (calculated octanol-water partition coefficient), the absence of stereocenters, and a larger radius. In contrast, the InhA and FabI inhibitors showed higher values for SlogP and NumAromaticRings. However, no statistically significant difference in AMW (average molecular weight) was observed compared to other enzyme datasets.

When comparing the physicochemical properties of active and inactive compounds within each enzyme dataset, clear differences were observed for FabI and InhA inhibitors. In the case of InhA inhibitors, active compounds exhibited statistically higher values for features such as AMW, HBD, TPSA, NRB, Bertz CT complexity, and sum of atomic polarizabilities (apol) compared to inactive compounds. Similarly, active FabI inhibitors showed significant differences with increased values in AMW, number of rings, NRB, BertzCT complexity, and sum of atomic polarizabilities (apol). One factor contributing to these variations is the inherent structure of the active sites, which influences parameters such as AMW, the number of rings, and HBD. In addition, the higher values observed are consistent with the typical progression of compounds undergoing further development, where the molecular mass and number of rings tend to increase in order to optimize the properties of the drug [22]. Interestingly, no significant difference was observed for SlogP, possibly indicating that achieving higher lipophilicity within the limits of Lipinski’s rules [23,24] is challenging, as the SlogP value of these compounds is already high in all datasets, reflecting the high hydrophobicity of the active sites in these enzymes. However, the increase in atomic apol presents a unique explanatory challenge that warrants further investigation.

The results of the statistical evaluation and Figure 5 emphasize the importance of these features in determining the activity of the compounds. It is noteworthy that the FabV and FabK inhibitors showed no statistically significant differences in the physicochemical properties assessed compared to inactive compounds. This lack of significance may be attributed to the relatively small dataset, as larger sample sizes are generally more likely to reject the null hypothesis [25]. However, from Figure 5 we can see that the active compounds in the FabV dataset have a higher HBD, HBA and TPSA than the inactive compounds in this dataset.

Next, we wanted to analyze the changes over time in the physicochemical properties of the ENR inhibitors to see if there was a trend that could be analyzed. However, no specific pattern was observed and the results are shown in Appendix A.

### 2.3. Lipinski’s Rule of Five

Bioactivity was evaluated in terms of drug-likeness characteristics, which is usually determined using Lipinski’s Rule of Five (also known as Pfizer’s Rule of Five or simply Rule of Five). This rule states that molecules with a molecular weight below 500 g/mol, no more than five hydrogen bond donors, no more than 10 hydrogen bond acceptors, and a logP value below 5 are generally considered drug-like [23,24]. While Lipinski noted that orally bioavailable active pharmaceutical ingredients (APIs) often meet these criteria, it is important to recognize that these guidelines should not be the sole determinants in the medicinal chemistry development of all small molecules. To illustrate, only half of all FDA-approved small-molecule drugs are both orally administrable and meet the Rule of Five [26]. Overly dogmatic adherence to Lipinski’s rules can lead to the exclusion of potential lead molecules that would otherwise be good drug candidates by overemphasizing oral bioavailability and excluding natural products, including antibiotics, most of which do not meet the Rule of Five criteria [27]. Furthermore, while oral administration is a desirable property of a drug, it is not a mandatory requirement [28].

When the inhibitors of the enzymes of interest were screened using Lipinski’s five-criteria rule, it was found that the majority of active compounds as well as inactive compounds conformed to Lipinski’s rules. Notable exceptions were found for FabK inhibitors, which had a non-compliance rate of 19%. For most compounds associated with single enzymes, the frequency of a single violation was less than 10% and the frequency of two violations was less than 7%. Remarkably, inhibitors targeting FabV, both in their active and inactive forms, had no Lipinski violations. This absence could be attributed to the relatively small dataset (Figure 6, Appendix A).

### 2.4. PAINS and Brenk Filters

Secondly, the dataset was analyzed for the presence of Pan-Assay Interference Compounds (PAINS) and unwanted functional groups using the Brenk filter. The PAINS filter identifies functional groups previously identified as false positives in other assays, which necessitated a thorough check for these substructures in our dataset [19,29]. Similar to the Lipinski rules, caution should be exercised when using PAINS filters, as a significant number of FDA-approved small-molecule drugs contain PAINS warnings [30]. As shown in Table 1 for the enzyme FabI, 11% of all inhibitors fell through the PAINS filter, with only 7% of all compounds being active and containing PAINS structures. In the case of the enzyme InhA, 8% of all inhibitors failed the PAINS filter, and only 1% of all compounds were active and contained PAINS structures. Analysis of FabK and FabV enzymes revealed that FabK inhibitors had no PAINS structures, while 20% of all FabV inhibitors had PAINS structures, with 10% being active inhibitors. Additional data on compounds classified as PAINS or Brenk can be found in the Appendix A.

In addition, the dataset was examined for the presence of undesirable components as described by Brenk et al. (Brenk filter) [31], e.g., substructures associated with toxicity and high reactivity. The results, presented in Table 1, show that a considerable number of compounds did not pass the Brenk filter. The table showing the most common unwanted groups found in ENR inhibitors (PAINS and Brenk) can be found in the Appendix A. It shows that in most cases undesirable groups were more abundant (50% or more) in the active forms. The most commonly encountered problematic functional groups in these compounds include Michael acceptors, oxygen-nitrogen single bonds, aliphatic long chains, nitro groups, thiocarbonyl groups, and imines.

Given the above considerations, one could conclude that applying PAINS and Brenk filters and then analyzing data without unwanted PAINS and Brenk structures would result in a significantly reduced dataset. The combination of structures that remain after the application of PAINS and Brenk filters resulted in only 746 molecules (51%; Appendix A). Therefore, it is advisable not to exclude molecules based solely on PAINS, Brenk, or even the previously mentioned Lipinski violation filtering.

### 2.5. Structural Diversity

The structural diversity of all four inhibitor datasets was assessed using four different representations: chemotype diversity, molecular similarity, t-SNE analysis, molecular complexity, and cluster analysis. The use of these different calculations provides a comprehensive representation of the structural diversity within all four enzyme libraries.

#### 2.5.1. Chemotype Diversity

The study of molecular scaffolds is of great importance as it facilitates subsequent steps in the drug discovery pipeline, including scaffold hopping [32]. Our focus was on identifying Murcos scaffolds to assess their diversity, resulting in 279 unique scaffolds across all four datasets [33]. Table 2 provides statistical data on the number of scaffolds (N), the proportion of scaffolds to the number of active molecules (N/M), and the number of singletons (N_sing_) for the four enzyme datasets described here. In terms of the proportion of scaffolds per molecule (N/M), the InhA library had the highest N/M count, and therefore the highest scaffold diversity, although the differences between InhA, FabI, and FabK were only marginal. Fab V, the smallest library, had only two scaffolds with the lowest N/M ratio, indicating the least diversity. These metrics provide valuable insight into scaffold diversity. However, it is important to note that they do not provide detailed information about the specific distribution patterns of these scaffolds within the datasets.

Among the techniques that measure the global diversity of a compound dataset are cyclic system recovery (CSR) curves that quantify the distribution of scaffold diversity [34]. In these diagrams, the closer the CSR line is to the diagonal, the greater the scaffold diversity. The plots in Figure 7 show that the chemotypes from the InhA database have only slightly lower scaffold diversity compared to those from the FabI database. Although CSR curves have previously been used effectively to analyze larger datasets [34], unfortunately the number of unique scaffolds in the FabV and FabK datasets was too small to produce a curve suitable for comparison, as the values for both curves did not reach 0 and will thus not be discussed in the following paragraph.

The following metric was derived from the CSR curves: the area under the curve (AUC) and the fraction of chemotypes required to find 50% of the molecules (F_50_), both shown in Table 2. The use of both metrics has been used successfully in previous studies as a measure of scaffold diversity because it is suitable for comparing collections of different sizes due to its independence of library size [6,11,12,13]. Higher F_50_ values and lower AUC values indicate greater diversity and vice versa. According to the F_50_ metric, the scaffolds in the InhA dataset had a slightly higher diversity than those in FabI, with F_50_ values of 0.158 and 0.133, respectively. As expected, the scaffold diversity of the InhA dataset was only slightly higher than that of FabI when both metrics, AUC and F_50_, were considered.

#### 2.5.2. Molecular Similarity

Tanimoto coefficients, also known as Tanimoto similarities, were calculated for MACCS (nbits = 166) fingerprints to assess the structural similarity between inhibitors in each dataset. The statistical values of pairwise intra-dataset Tanimoto similarity for each enzyme are shown in Appendix A, and the intra- and inter-group similarity results are shown in the matrix in Figure 8 and in the Appendix A.

The analysis revealed limited diversity in the FabK and FabV inhibitors (with a within-group similarity of 0.70 and 0.74, respectively), as expected due to the smaller scale and synthesis by only a couple of research groups for both datasets. In addition, the two pairs of inhibitors in these datasets were much more similar to each other than to the inhibitors of the other two enzyme groups. In contrast, InhA and FabI exhibited greater diversity, with within-group similarities comparable to between-group similarities for all other inhibitor groups. Remarkably, both the InhA and FabI datasets showed at least a 10-fold increase in the number of compounds, emphasizing that the largest dataset is not necessarily 10 times more diverse. This is clearly illustrated here, where the differences between the mean similarities are not significantly larger.

#### 2.5.3. t-SNE Analysis

The visualization of chemical space is crucial to identify whether the datasets occupy the same chemical space [19]. In Figure 9, t-SNE analysis using MACCS fingerprints illustrates characteristic substructure patterns for each enzyme dataset. The graph representing all four enzymes shows that there is no overlap between the datasets and each enzyme occupies its own place in chemical space. The number of clusters is directly proportional to the number of compounds; the greater the number of compounds, the greater the number of clusters. The diagram depicting all four enzymes shows a higher structural diversity of the InhA dataset compared to the FabI dataset, which is reflected in a higher number of clusters. We can also observe how small and structurally less diverse the FabV and FabK datasets are, as the chemical space they occupy in the diagram is very limited.

In addition, Figure 9 also illustrates the differences in structural diversity between active and inactive compounds for each enzyme separately. In particular, the diagram for the enzyme InhA shows a greater chemical diversity in the inactive compounds, which can be seen in several clusters. This indicates a broader spectrum of chemical structures among the inactive inhibitors for InhA. On the other hand, FabI shows a slightly larger number of clusters for active compounds, indicating diverse chemical profiles among the active inhibitors. Unfortunately, insufficient data for the enzymes FabV and FabK prevent the formation of clear clusters.

#### 2.5.4. Complexity of Datasets

Molecular complexity is an important aspect of drug design as it is positively associated with target selectivity [17,18] and consequently with toxicological profiles [34] and success in development in the clinic [35]. Various complexity metrics are commonly used, including topological, physicochemical, and structure-based approaches, as well as graph-theoretical methods [36]. For our analysis, we used the BertzCT complexity index as a measure of complexity, as it is specifically designed to provide a more comprehensive estimate of complexity by taking into account its molecular graphs and incorporating information about the complexity of bonds and heteroatoms [19,37]. A correlation between pIC_50_ and the BertzCT index is presented in Figure 10.

Although previous studies have found a correlation between pIC_50_ and the BertzCT index [10], in our case we only observed a weak correlation for the InhA dataset, with a Pearson correlation coefficient of 0.340 for active compounds and −0.107 for inactive compounds (*p* < 0.05). No obvious correlation was observed for other three enzyme datasets (Appendix A).

Nevertheless, we found a difference in complexity when only activity is considered. There was a statistically significant difference between active and inactive compounds in InhA and FabI inhibitors, with active compounds exhibiting higher complexity. However, it is noteworthy that this observed increase in complexity could be related to the development pathway that compounds take in research projects. As compounds are identified, they are often further developed, leading to a focus on synthesizing larger and more complex molecules. Consequently, the observed increase in complexity during a research project can be attributed to the preference for synthesizing larger compounds rather than the intrinsic complexity of the synthetic pathway itself [38].

Nonetheless, we also observed differences in complexity between inhibitors in all four datasets, although the differences were small. We found that InhA (BertzCT = 1004) and FabK (BertzCT = 1090) datasets had a slightly higher complexity than the FabI dataset (BertzCT = 977) and a much higher complexity than the FabV dataset (BertzCT = 660).

#### 2.5.5. Analysis of Clusters

To better understand and quantify the molecular diversity within each enzyme dataset and identify significant structural classes associated with their activity, we performed clustering using ECFP4 molecular fingerprints that capture more detailed molecular substructures than MACCS fingerprints. A threshold of 0.75 for Tanimoto similarity was manually adjusted to allow spontaneous generation of clusters with an optimal balance between too small and too cohesive clusters (high intra-cluster similarity) and too large and diverse clusters (low intra-cluster similarity). The total number of clusters in each dataset served as a metric for chemical diversity.

This analysis identified a total of 226 clusters for all datasets together, a considerable number considering the 1412 compounds. This result also highlights the diversity within particularly large clusters, such as InhA and FabI. Most of these clusters were small, comprising only one or two compounds. In addition, about 60% of the clusters in the InhA, FabK and FabI datasets consisted of fewer than 10 compounds. The main statistical data for all four datasets, with particular emphasis on the five largest clusters of each dataset, are shown in Table 3, while their structures are shown in Figure 11.

The number of clusters in each dataset shows a correlation with the respective dataset size. InhA, which was the largest dataset, had 77 clusters, while FabV, which was the smallest, defined only one cluster. When comparing the number of compounds that contained only active or inactive compounds, a clear trend emerged: the number of completely inactive clusters clearly exceeded the number of completely active clusters. Only 9% of the compounds in the InhA and FabI datasets were exclusively active, a significantly lower percentage than the 69% in InhA and 31% in FabI, which were exclusively inactive. Remarkably, these inactive clusters tended to be small, with over 60% containing only one compound per cluster. Similar to the t-SNE analysis, it is evident that the diversity of compounds categorized as inactive or of low activity was much greater compared to the active compounds.

It is important to emphasize that the definition of inactivity was stringent, as most compounds in these inactive clusters had pIC_50_ values above 5.5. Nonetheless, the significant number of inactive clusters underscores the thorough investigation of structure–activity relationships (SAR) in the InhA and FabI datasets, which led to the identification of compounds with nanomolar activities as well as those with weak or no activity.

For a comprehensive evaluation of structural diversity within each dataset, the number of clusters was standardized by the total number of compounds in each dataset. Based on this criterion, the FabI dataset appears to be less diverse than the InhA dataset. Further scrutiny of the physicochemical properties of the most populated clusters, as depicted in Appendix A, revealed that the characteristics of most clusters aligned with the observations made on the combined dataset. The majority of compounds demonstrated lipophilicity between 3 and 6 and a molecular weight ranging from 300 to 550 (Appendix A). However, an exception was observed in cluster 4 of the InhA dataset, where pyrrolidine-based derivatives displayed a higher average molecular weight (Mw 555), a lower average lipophilicity (logP 1.3), and a higher number of hydrogen bond acceptors (HBA 7.4).

Examination of the clusters with the highest number of compounds reveals the development trends observed in both academic and industrial settings over the last two decades. Despite comprehensive reviews of inhibitors for all four enzymes over the years, it is considered crucial to quickly highlight certain historical contexts and fundamental properties of the most important classes.

The predominant cluster in all four datasets was cluster 1 of the FabI dataset which included mainly acrylamides. This cluster comprised 208 compounds, accounting for 47% of the total FabI dataset, with a mean intra-cluster similarity of 0.3. Remarkably, 77% of the molecules in this cluster were active (Table 3). While the central pyridine acrylamide structure is common to all compounds in this cluster, their diversity arises from the different substituents on both sides of the molecules. In addition, it includes some of the most potent compounds among all FabI inhibitors, with activities ranging from above 5.4 log units and pIC_50_ to picomolar activity.

The identification of acrylamides as FabI inhibitors dates back to a high-throughput screening (HTS) campaign initiated by SmithKline Beecham in the early 21st century. In the following two decades, the development of these compounds was continued by several companies, including SmithKline Beecham, Affinium Pharmaceuticals, Vitas Pharma Research Private Limited (India), FAB Pharma, Janssen R&D, and Aurigen Discovery Technologies. Despite initial concerns about excessive patenting and a perceived lack of novelty within this structural class, our analysis shows that this class is comparable to or even more diverse than other classes of inhibitors when considering the similarities within a cluster [6].

The second largest cluster, cluster 1 of InhA inhibitors, shows structures containing oxopyrrolidine-3-carboxamide, piperidine carbonyl, and pyperazine carbonyl units. This cluster comprises 162 compounds, accounting for 16% of the total InhA dataset (Table 3). Approximately 39% of the molecules in cluster 1 exhibit activity and show a mean intra-cluster similarity of 0.28. The diversity of these molecules is achieved by highly lipophilic aromatic or carbocyclic substituents on both sides of the pyrrolidine ring. Their activities range from submicromolar to low micromolar concentrations and cover a range of 3.2 log units.

The third largest and particularly significant group comprises diaryl ethers identified in cluster 2 of the InhA dataset, cluster 2 and 3 of the FabI dataset and cluster 1 of the FabV dataset, totaling 114, 59, 40, and 15 compounds, respectively. Compared to other analyzed clusters, these compound clusters show a high diversity, with the cluster similarity ranging from 0.4 (FabV, cluster 1) to 0.52 (InhA, cluster 2). Triclosan serves as a prototype molecule for these clusters, which was originally thought to exert antibacterial activity through non-specific destruction of bacterial cell walls. In 2004, Sivaraman et al. [39] reported that triclosan is a slow, tight-binding inhibitor of *E. coli* and *S. aureus* FabI that binds to the NADP product complex with a Ki of 7 pM. Despite its modest antibacterial activity (MIC = 12.5 μg/mL) and unfavorable properties such as low water solubility, ecotoxicity, and potential disruption of thyroid and sex hormone homeostasis [20], it is still considered a viable lead compound for further development.

Analogs of triclosan synthesized in recent decades exhibit enhanced inhibitory activity and may be present in the analyzed clusters. The most potent compounds of this class identified in the FabI and InhA datasets have shown sufficient minimum inhibitory concentrations (MIC) against strains resistant to triclosan. In quantitative structure–activity relationships (QSAR), the lipophilicity of the alkyl chain was found to significantly influence activity. Compounds with increased activity against InhA are observed when lipophilicity is increased, with a C5 chain on the phenyl ring in particular proving most effective. However, high lipophilicity poses a challenge for pharmacokinetics as the most potent compounds have logP values above 5. Efforts have been made to introduce hydrophilic substituents, but unfortunately these have not been able to inhibit InhA in the nanomolar range. However, a few inhibitors with a modified B-ring, in which phenyl was replaced by pyridine, showed reduced logP and activity in the nanomolar range and were well tolerated [40].

### 2.6. Matched Molecular Pairs (MMP)

One of the most popular methods for determining how specific structural changes affect physicochemical or biological properties is MMP (Matched Molecular Pair) analysis. MMP is defined as a pair of compounds that differ only at a single site and are characterized by a specific substructure. A key feature of MMP is that the compounds in this pair are linked by a well-defined transformation [41]. The strength of this method lies in its simplicity and its ability to quickly predict differences in activity or property values rather than determining exact values. MMP, which focuses on local transformations, is used to predict the properties of compounds, recommend the preparation of the next compound, and identify the changes with the greatest impact (activity cliffs) and the least impact (bioisosteres) on activity [42].

Our analysis show that both core (number of cuts > 1) and end (number of cuts = 1) transformations are prevalent in the conspicuous transformations in the InhA and FabI datasets (Appendix A). In particular, we found cases where a single pair could include multiple transformation types, each specifying a different context. This diversity, even if formally occurring only once, stems from simpler and more frequently occurring transformations. As expected, the most common transformations primarily resulted in minor structural changes (Appendix A). A significant proportion involved substitutions between hydrophobic groups, such as the exchange of a methyl group or a halogen atom with another halogen atom and vice versa. This is in line with expectations, given the relatively hydrophobic nature of the binding sites of all four enzymes [2]. In addition, CH_2_ groups were frequently inserted or removed and ring substitution patterns were altered, which is particularly noticeable in the FabI and InhA datasets. Most of these transformations involved functional groups within ring systems. Interestingly, the effect of each transformation on activity varied depending on the substitution pattern. For example, the consequences of the transformation of a methyl substituent in one ring system were not necessarily repeated in another ring with a different substitution pattern.

A notable observation when analyzing the results of the MMPs is the presence of activity cliffs. This analysis can shed light on which types of transformations are more likely to lead to a significant increase or decrease in activity. However, our results suggest that the most common structural transformations rarely led to significant changes in activity, as activity cliffs were rarely observed in these common transformations. Although all of these transformations were found to be significant, the average activity gain per transformation was less than 0.5 of ΔpIC_50_ in almost all cases. This indicates that most of the structural changes of these inhibitors resulted in linear but small changes in activity.

We then examined the transformations that most frequently lead to activity cliffs. To categorize the transformations based on their impact on activity, we assigned each transformation to one of four categories based on the following criteria:Activity cliff—At least one of the compounds in the MMP is active and the difference in activity is at least 100 nM.Soft cliff—At least one of the compounds in the MMP is active and the difference in activity is less than 100 nM.Similarly active—Both compounds in the MMP are active and the difference in activity is less than 100 nM.Similarly inactive—Both compounds in the MMP are inactive and the difference in activity is less than 100 nM.

The distribution of the individual categories within the enzyme dataset is shown in Figure 12. Due to the considerable differences in the number of transformations within the individual datasets, it is difficult to visualize them directly. Therefore, we converted the counts into percentages of all transformations per enzyme. Of the total 7448 transformations, only 576 were identified and categorized as activity cliffs, representing less than 8% of all transformations. In particular, only 3% and 8% of activity cliffs were found in the FabI and InhA datasets, respectively. This suggests that navigating the chemical space of InhA is slightly more difficult than in FabI. Nevertheless, analyzing the occurrence of activity cliffs is crucial for understanding which transformations are more likely to lead to undesired or desired activity shifts.

To obtain representative results regarding the transformations that lead more frequently to activity cliffs, we analyzed only the cliffs that occurred in at least 10 pairs per transformation and in at least 5% of the pairs per transformation. Under these conditions, we identified 20, 2, and 252 transformations for the FabI, FabV, and InhA datasets, respectively, that met these criteria. No transformations meeting these criteria were identified in the FabK dataset. Representative transformations for the FabI and InhA datasets are shown in Figure 13, and the transformations with the highest frequency of activity cliffs in each dataset are listed in the Appendix A. The transformations with the highest proportion of activity cliffs involve more significant structural changes than those observed in the most frequent transformations. Changes in the substitution pattern and the addition or deletion of a CH_2_ group were prevalent in all datasets. In FabI, an activity cliff is often associated with the complete transformation of one carbo- or heterocyclic cyclic system into another and the transformation of a polar functional group (e.g., OH) into a more lipophilic group (e.g., OCH_3_). In InhA, activity cliffs were also associated with the conversion of one cyclic system into another; however, in most cases, one lipophilic carbocycle was transformed into another carbocycle. We also found that the highest number of activity cliffs analyzed corresponded to compounds from clusters 1 and 5 of the FabI dataset and clusters 1, 2, 3, and 4 of the InhA dataset.

While it is difficult to completely avoid compounds that exhibit activity cliffs in QSAR analysis, certain scaffolds and fragments often show a predisposition for their occurrence. Predicting which transformations are more likely to exhibit activity cliffs is challenging, especially given the relatively low proportion of activity cliffs per transformation. Figure 14 shows the fragments that are more frequently associated with activity cliffs in ENR datasets. The corresponding physicochemical properties can be found in the Appendix A. We identified 574, 302, six, and three unique fragments for the InhA, FabI, FabK, and FabV datasets, respectively. It is evident that the fragments most abundant in FabI pairs contain a higher number of heterocyclic scaffolds than the fragments in InhA transformations, resulting in a slightly higher median logP for the InhA dataset (2.0 vs. 1.8). As expected, only a limited number of fragments were identified for the FabK and FabV datasets.

While MMP analysis provides valuable insight into transformations that often lead to non-linear changes in activity, it was a challenge to extract useful information for the FabK and FabV datasets. Despite the diverse chemical space, there is still a significant number of compound pairs that are intriguing and warrant further investigation to understand ENR inhibitory activity.

### 2.7. SHAP Analysis

SHapley Additive exPlanations (SHAP) was used to analyze the influence of functional groups on the compounds in each dataset. SHAP provides ratings of the importance of individual features, taking into account the interactions between features, based on the Shapley values from cooperative game theory. A 307-bit substructure fingerprint was generated using Padel 2.21 software [43], and Random Forest (RF) was selected for the SHAP analysis, as it outperformed all other algorithms tested (Appendix A). Due to poor performance in our model, FabV was excluded from the analyses. Figure 15 shows the results, with the *x*-axis representing the SHAP values and the *y*-axis representing the selected features. Each point in the graph represents a SHAP value for a predictive feature, with red indicating a higher value and blue a lower value. The distribution of the red and blue dots provides information about the overall influence of the individual characteristics on the result of the model.

In the FabI dataset, our observations suggest that an increase in the number of Michael receptors, alkene groups, and tertiary amide groups together with a decrease in the number of secondary carbon, secondary amide, basic *N*-heterocyclic rings, and carbonic acid derivatives contribute to an increased probability of activity in an RF model. Conversely, a reduction in the number of these features leads to a lower probability of activity in FabI. For FabK, our results suggest that increasing the number of aromatic groups, including *S*- and *N*-heterocyclic systems, as well as the number of carboxylic acid derivatives such as esters or amides and CONS bonds, increases the probability of activity in FabK within an RF model. Furthermore, we find that increasing the number of basic *N*-heterocyclic groups has a stronger effect on activity than increasing the number of non-basic *N*-heterocycles.

With respect to InhA, our results indicate that increasing the number of 1,3-tautomerizable groups, secondary amides and aromatic rings while decreasing the number of basic *N*-heterocycles, carboxylic acid derivatives and alkenes leads to an increased probability of activity on InhA in an RF model. A reduction in the number of secondary carbons also contributes to an increased probability of activity. In all datasets, most features have SHAP values between −0.2 and 0.2, indicating that their influence on the overall prediction is limited. This illustrates the complexity of our model, where predictions are highly dependent on the intricate interactions between multiple traits.

Both the MMP and SHAP analyses revealed common transformations that are directed by ligand–target interactions and emphasized the importance of specific functional groups, particularly in the InhA and FabI datasets. These transformations, which include core and end changes, reflect the dynamic nature of ligand binding and the structural plasticity of binding sites. This is supported by crystal structures showing similar binding modes for most inhibitors [44,45]. Remarkably, the most frequent transformations were primarily substitutions between hydrophobic and electron-withdrawing groups, consistent with the hydrophobic nature of the enzyme binding sites [46]. Previous studies on triclosan and its analogs emphasized the preference for small substituents on the B-ring to avoid steric hindrances and unfavorable interactions within the active site, which is consistent with our results. In addition, experimental studies on specific inhibitors, such as 5-chloro-2-phenoxyphenol, have highlighted the importance of certain structural features, such as methyl groups, in increasing binding affinity and influencing retention time in the active site [47,48,49,50,51]. Increasing the polarity of both rings of triclosan, e.g., by replacing the B-ring with isosteric *N*-heterocycles, resulted in decreased activity of pyridine analogs and almost complete loss of activity of pyrimidine analogs. In addition, the introduction of polar groups on the B-ring system, such as carboxylic acid, primary amide and tetrazole, led to a significant decrease in activity. Similar trends were observed with other inhibitor classes such as pyrrole carboxamides and acrylamide analogs, where lipophilic, electron-withdrawing substituents in positions 3 and 5, preferably Cl, Br, or CF_3_, proved to be particularly advantageous [48]. Nevertheless, some electronegative residues, such as the hydroxyl group in triclosan, are crucial for the formation of hydrogen bonds with Tyr 158 hydroxyl, with some notable exceptions, such as thiazole inhibitors. This hydrogen bonding network also extends to the hydroxyl group of NAD+ ribose, with the electronegative group typically located in close proximity to a cyclic unsaturated or aromatic system that interacts with the pyridine of a cofactor via stacking [11]. Given the dynamic nature of InhA inhibition and its structural flexibility, it has been suggested that consideration of different inhibitory pathways may be necessary for effective inhibition [44]. Integrating these results from computational analysis with experimental data will provide a comprehensive understanding of the dynamics of ligand–target interactions and thus inform rational drug design strategies to develop new inhibitors with improved potency and selectivity.

## 3. Materials and Methods

### 3.1. Data Collection, Preprocessing, and Classification

A dataset of 3743 molecules was compiled from publicly available literature, ChEMBL [52], BindingDB [53], and our database until April 2023. The literature data were manually extracted by drawing each molecule in ChemDraw (version 22.0.0), and then the SMILES codes were transferred into an Excel format along with the IC_50_ values. First, the data from each database were prepared using the KNIME Analytics Platform 5.0 [54], using a modified workflow derived from the TeachOpenCADD pipeline (https://hub.knime.com/volkamerlab/space/TeachOpenCADD (accessed on 28 February 2024)). Data from all four sources were then consolidated. Molecules for which no relevant data were available (SMILES, enzymes) were excluded from further analysis. Duplicate values were eliminated, leaving only entries with higher activity. In cases where IC_50_ data were missing or indicated inactivity of the substance (activity of the inhibited enzyme greater than 80% or inhibition less than 20%), an IC_50_ value of 100 μM was set. Furthermore, inhibitors specific for Toxoplasma and Plasmodium were excluded, which corresponds to the focus of this research on the inhibition of bacterial enzymes.

Finally, we converted the IC_50_ values into pIC_50_ values and categorized the molecules as active or inactive based on a defined threshold. Compounds with a pIC_50_ value greater than or equal to 5.5 were considered active inhibitors, while compounds with a pIC_50_ value less than 5.5 were considered inactive inhibitors. The cut-off value for activity was set between 5 and 6, which led us to choose an average value of 5.5. Finally, we obtained a dataset of 1412 molecules (Appendix A) with which we performed additional analyses and data manipulations.

### 3.2. Calculation of Molecular Descriptors

Subsequently, RDKit descriptors were generated using the RDKit Descriptor Calculation node implemented in KNIME Analytics Platform 5.0 [54]. A total of 39 features were generated, including enhanced/hybrid logP (SlogP), total polar surface area (TPSA), average molar weight (AMW), number of rotatable bonds (NumRotatableBonds), number of hydrogen bond donors (NumHBD), number of hydrogen bond acceptors (NumHBA), fraction of carbon atoms that are sp3 hybridized (FractionCSP3), and many others. The BertzCT descriptor was calculated using Mordred, a descriptor-calculation software module implemented in Python 3.7.4 [55]. Additionally, the substructure count fingerprint was calculated using the PaDEL-descriptor software [43].

### 3.3. Lipinski’s Rule of Five, PAINS, and Brenk

The dataset was analyzed in KNIME Analytics Platform 5.0 [54], according to Lipinski’s Rule of Five, and the number of Lipinski violations was calculated. Finally, the dataset underwent analysis for unwanted substructures, including those classified as PAINS and Brenk (Appendix A).

### 3.4. Normal Distribution Testing

Before the features were compared between the two activity classes, a normality check was carried out. Continuous features were tested for normality using the Kolmogorov–Smirnov and D’Agostino–Pearson tests. The results were then adjusted using the Holm–Bonferroni method (*p* < 0.05). Individually, BertzCT, fragment complexity (fragCpx), log S by filter-it (FilterItLogS), apol (sum of atomic polarizabilities), Fraction of rotatable bonds—excluding terminal bonds (RotBFrac), SlogP, TPSA, and average molecular weight (AMW) were compared for each enzyme (both in active and inactive forms). A collective assay was then performed for all active and all inactive forms [56].

It was found that none of the features tested for both the active and inactive datasets individually, as well as for the enzymes as a whole, showed a normal distribution according to the Kolmogorov–Smirnov test. However, certain descriptors were identified as normally distributed using the D’Agostino–Pearson test (Appendix A). It is important to note that non-parametric tests are less powerful than parametric tests when analyzing normally distributed data. This means that a larger sample size is required to reject the null hypothesis in non-parametric tests compared to parametric tests. Given the small sample size in some cases (e.g., FabK and FabV datasets), we treated all data as non-normally distributed. Histograms can be found in the Appendix A for further reference (Appendix A) [57].

### 3.5. Mann–Whitney U Rank Test

The Kruskal–Wallis H test was first used to test which features differed in all four enzyme datasets [56]. Subsequently, samples that showed statistical differences (*p* < 0.05) were subjected to additional tests to determine differences between individual samples using the Mann–Whitney U test [57]. The *p*-values for the Mann–Whitney U-rank test were calculated for each enzyme by comparing its active and inactive forms across various molecular descriptors using Python scripts with the imported modules statsmodels and scipy. The analysis also included the application of Holm–Bonferroni corrections [58].

### 3.6. Visualization Using t-SNE Analysis

To analyze differences between enzymes and their active and inactive compounds, a multidimensional space was converted into 2D using t-SNE for visual clarity [59]. The analysis, conducted with Python’s scikit-learn module [60] using fingerprints generated with PaDEL [43], Modred [55], and RDKit [21], initially considered multiple fingerprints, but ultimately, only the MACCS fingerprint was included in the research paper. This decision was based on a thorough analysis that revealed MACCS provided the best visual separation of data among the considered fingerprints.

### 3.7. Visualization Using t-SNE Analysis

To analyze the differences between enzymes and their active and inactive compounds, a multidimensional space was converted to 2D using t-SNE to increase visual clarity. Several fingerprints were initially considered in the analysis, which was performed with the scikit-learn module of Python using fingerprints created with PaDEL, Modred, and RDKit, but ultimately only the MACCS fingerprint calculated with RDKit was included in the research. This decision was based on a thorough analysis, which revealed that MACCS offered the best visual separation of the data among the fingerprints considered [19].

### 3.8. Clustering Analysis

Clustering of datasets to create ‘structural families’ was performed using Python in a Jupyter Notebook environment. First, SMILES representations of the compounds were used to calculate ECFP4 molecular fingerprints. A distance matrix was then created to assess the structural similarities between the compounds using Tanimoto similarity as a metric. The Taylor–Butina clustering algorithm [61] was then applied to create clusters based on a specific threshold. Different cut-off values for Tanimoto similarity were examined, ranging from 0.1 to 0.9. The resulting clusters were visually examined to strike an optimal balance between clusters that were too small and overly cohesive (high intracluster similarity) and clusters that were too large and diverse (low intracluster similarity). After a comprehensive visual inspection of the results, an optimal cutoff value of 0.75 was chosen, which struck a balance between a sufficiently manageable number of clusters and a substantial number of clusters with moderate to high intra-cluster similarity. Finally, the results were summarized and systematically organized for further analysis, which was carried out KNIME Analytics Platform 5.0 [54]. The number of unique clusters and the distribution of compounds within these clusters were documented together with their respective biological activities.

#### 3.8.1. Maximum Common Substructure (MCS)

To further analyze the clusters, the Maximum Common Substructure (MCS) of each cluster was calculated in KNIME Analytics Platform 5.0 [54] using the RDKit MCS node. RingMatchesRingOnly was set to True to ensure that only ring atoms were matched with ring atoms when comparing different structures. In addition, the median of similarity within a cluster was calculated, which represents the median of similarities between every two compounds in a given cluster. Other parameters, such as the number of compounds in each cluster, the number of active compounds in a cluster, and the third quartile of the pIC_50_ values of the compounds in a cluster were also calculated.

Finally, the results were aggregated and systematically organized for further analysis. This process included documenting the number of unique clusters as well as the distribution of compounds within these clusters and their respective biological activities. This comprehensive approach gave us a deep understanding of both the structural relationships and the activity profiles of the compounds studied.

#### 3.8.2. Analysis of Scaffolds

The analysis of chemical scaffolds is a fundamental tool widely used in medicinal chemistry, cheminformatics, and chemogenomics. A scaffold can be defined as the central structural framework common to a group of molecules [62]. Compounds derived from bioactive scaffolds often exhibit pharmacological activity and utilize the same synthetic pathways, making their analysis an essential step in drug discovery. In addition, they play a crucial role in the identification of core structures that serve as a starting point for further structural optimization, in the exploration of SAR, in the synthesis of new compounds (scaffold hopping [32]), and in the generation of activity profiles representing drug series [63]. In this study, Bemis–Murcko (BM) scaffolds [33] as implemented in RDKit were used and analyzed in Python. These are defined as the core structure of a joint consisting of rings and links [21]. The scaffolds were extracted from the compounds by removing all substituents while retaining the ring systems and their connecting linkers. Subsequently, each unique scaffold was coded with a unique number, which was used to calculate the chemotype diversity.

### 3.9. Diversity of Chemotypes

For each of the four enzyme datasets, we performed an analysis to determine the number of unique chemotypes (N) and to identify chemotypes consisting of only one compound, referred to as singletons (N_sing_). BM scaffolds were first calculated for each molecule and then several key metrics were calculated. These included the proportions of chemotypes and singletons relative to the size of each individual dataset (M) and the proportions of singletons relative to the total number of chemotypes across all four datasets (N/M. N_sing_/M. and N_sing_/N).

Previous analyses of scaffold diversity have shown that cyclic system retrieval curves (CRS) [20,64,65,66,67] are an effective tool for quantifying the scaffold diversity of chemical libraries. To generate the CSR curves for each enzyme dataset, the number of unique scaffold compounds was calculated and sorted according to their frequency of occurrence in Python. The proportion of compounds was then plotted against the proportion of chemotypes contained in these compounds. From these CSR curves, the following metric was then determined: the AUC and the fraction of chemotypes required to find 50% of the molecules (F_50_).

In contrast to the CSR curves, which provide information about the scaffold diversity of all datasets, Shannon entropy (SE) is used to characterize the distribution (diversity) of compounds among the n most populated chemotypes. An SSE value closer to 1 indicates maximum scaffold diversity, while an SSE value closer to zero (0) indicates low scaffold diversity. In this study, we analyzed different values of n, ranging from 5 to 20. More detailed information on the calculation of SE and SSE can be found in references [64,65,66,67].

### 3.10. Molecule Similarity

Tanimoto coefficients (or Tanimoto similarities) were calculated for MACCS (nbits = 166) and ECFP4 fingerprints (radius = 2, nbits = 2048) using Python 3.7.4 to measure the structural similarities between the compounds in each dataset. Both MACCS and ECFP4 are binary fingerprints representing molecular structure. The difference between them is that MACCS encodes the presence or absence of certain substructures or features in a molecule, while ECFP4 is a circular fingerprint that encodes information about the atomic and bonding environments around each atom in the molecule.

Pairwise similarity matrices were calculated for each dataset, ranging from 0 (least similar) to 1 (most similar). Compounds were ranked within each enzyme based on mean similarity scores, and Cumulative Distribution Function (CDF) diagrams were generated [68]. These diagrams illustrate the similarity of chemical structures within and between each enzyme group. The *x*-axis represents the pairwise similarity values, while the y-axis represents the cumulative distribution function. Each enzyme is represented by a separate line in the graph.

In addition, both intramolecular and intermolecular similarities were calculated, which form the basis for creating a heat map of similarities. Various statistics, including mean, median, standard deviation, maximum, minimum, and quartiles, were also calculated for the pairwise similarities.

### 3.11. SHAP Analysis

Using Padel 2.21 software, a 307-bit substructure fingerprint was generated for each compound in all four datasets, with each bit representing a corresponding number of functional groups [43]. To avoid data leakage, the dataset was divided into training and test sets by stratification in a 7:3 ratio. All subsequent data manipulations were carried out exclusively on the training dataset. Invariant and highly correlated bits (Spearman correlation coefficient > 0.7) were eliminated, resulting in a refined dataset with 78 bits. It is important to emphasize that our main focus was on analyzing the importance of features and not on developing a model for large-scale screening. Nevertheless, it was crucial to obtain robust test results across different datasets. To overcome this challenge, we performed strategic oversampling of the FabK and FabV classes during training using the Synthetic Minority Oversampling Technique (SMOTE) to improve the model’s ability to recognize patterns in these minority classes. In addition, various feature selection techniques were used, including KBest, principal component analysis (PCA), Permutation Feature Importance (PFI), Select from Model (Selmodel), and Recursive Feature Elimination (rfe). Each generated dataset was evaluated further with different classification models, including Random Forest, XGBoost, Extra Trees, LightGBM, Decision Tree, Gradient Boosting, AdaBoost, K-Nearest Neighbors, Support Vector Classifier (SVC), and Linear Discriminant Analysis (LDA). The optimization of the hyperparameters was performed with GridSearchCV, whereby a nested cross-validation with the F1 macro was implemented as an evaluation criterion. This metrics calculates the F1 score for each class and then calculates the average of these scores, assigning the same weight to each class, regardless of the number of instances in the class. For the inner cross-validation, five splits were used, while 10 splits were used for the outer cross-validation. Normalization of the data was performed exclusively with the MinMax scaler for SVC and LDA classifiers which required normalization; otherwise, no normalization was performed.

The best performing classification model was then used to analyze Important features to predict the activity of each dataset using SHAP (sHapley Additive exPlanations). SHAP is a game-theoretic approach developed to explain the output of any machine learning process [69]. Evaluation metrics such as accuracy, precision, and F1 macro were used to evaluate the effectiveness of the model. Despite efforts to mitigate overfitting, slight overfitting of the Random Forest model was observed. For example, the F1-macro cross-validation score averaged 0.956 on the training set and 0.769 on the test set (Appendix A). Closer inspection revealed that the overfitting was primarily due to the poor performance of the model on the small FabV dataset, which significantly affected the overall score, as shown in Figure 15 and Appendix A. With precision and recall values of 0.5 for the FabV test dataset, our model was deemed unreliable for predictions on this dataset. Consequently, this dataset was excluded from further analysis.

### 3.12. Matched Molecular Pairs (MMP)

Matched Molecular Pairs (MMP) is an approach that deals with the question of how specific local chemical changes influence biological activity. It is used to predict the properties of compounds, recommend the next compound for synthesis, identify influential changes (activity cliffs), and assess minimal effects (bioisosteres) on activity [70]. Our focus was on the investigation of activity cliffs in four enzyme datasets.

To investigate this, we used the algorithm developed by Hussain and Rea [47] and implemented it in KNIME Analytics Platform Version 5.0 [54] with Vernalis nodes. In the initial phase, all compounds were fragmented using the MMP MoleculeFragment node, which identifies the key-value pairs of the fragmented molecules for direct use in a subsequent Fragments to MMPs node. Acyclic single bonds between two non-hydrogen atoms, along with their two- or three-bond combinations (referred to as the number of cuts), were deleted. The node allowed two cuts along a single bond, resulting in a single bond value. Incoming explicit hydrogen atoms were removed before fragmentation, and stereochemical considerations were not taken into account in this process. Pairs of similar molecules (Tc > 0.56) differing by no more than 10 heavy atoms were identified. Each pair was characterized by chemical transformations, considering only pairs with at least four common transformations. For each pair, the difference in pIC_50_ changes for each transformation. The obtained data was then further analyzed in Python 3.7.4.

## 4. Conclusions

In summary, our study provides a comprehensive investigation of small molecules targeting ENR enzymes and offers crucial insights into structure–activity relationships, physicochemical properties, and structural features. With a dataset comprising 1412 ENR inhibitors, this research lays a solid foundation for rational drug design, particularly in the context of developing new antimicrobial agents that incorporate novel scaffolds and chemical properties, with a particular focus on combating tuberculosis.

Using advanced cheminformatics tools, we have identified key physicochemical properties that influence bioactivity, such as lipophilicity, hydrogen bonding, and enzyme-specific structural features. In particular, a detailed comparison between InhA and FabI inhibitors revealed significant differences in lipophilicity, aromatic rings, and hydrogen bonding, providing valuable information for tailored drug discovery strategies. However, the FabK and FabV datasets showed limited diversity, highlighting the urgent need for increased research efforts to explore new structural classes for these enzymes.

Our complexity analysis and clustering study revealed trends in drug development pathways and showed a nuanced correlation between increased biological activity and greater complexity. The clustering analysis revealed development trends over the last two decades and highlighted the need for a strategic approach to drug development. In addition, the study of matched molecule pairs (MMPs) revealed activity cliffs and showed that frequent structural changes usually led to minor changes in activity.

The machine learning models and SHAP analysis revealed specific features that influence the probability of activity of each enzyme. In the FabI dataset, an increased number of Michael receptors, alkene groups and tertiary amide groups correlated with a higher probability of activity, while a decrease in these features led to a lower probability. In FabK, our results indicated that a specific increase in aromatic groups and certain chemical bonds increased the probability of activity.

Looking ahead, our bioinformatic analysis of chemical space is very promising for future ENR inhibitor discovery. The findings from the InhA and FabI datasets provide a solid foundation for the development of novel inhibitors, while the limited analyses of the FabK and FabV datasets emphasize the urgent need for diversified structural exploration. As computational methods evolve, the integration of these tools will undoubtedly play a crucial role in the development of novel ENR inhibitors with improved efficacy and safety profiles.

## Figures and Tables

**Figure 1 antibiotics-13-00252-f001:**
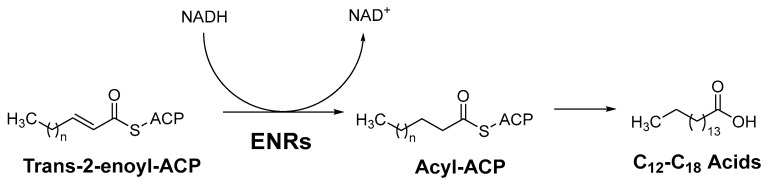
The last step of fatty acid elongation, ACP = acyl carrier protein, ENRs = Enoyl-acyl carrier protein reductases [2].

**Figure 2 antibiotics-13-00252-f002:**
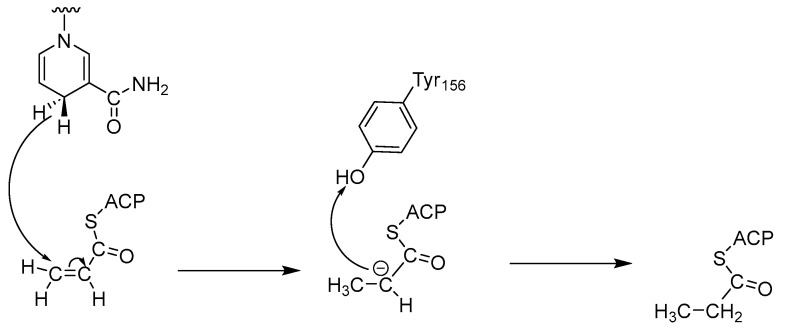
Mechanism of catalysis by ENRs [7].

**Figure 3 antibiotics-13-00252-f003:**
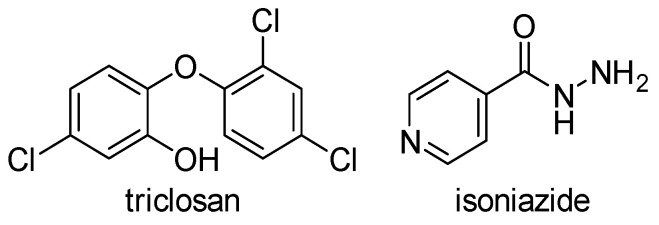
Inhibitors of ENRs in use today [2,10,11].

**Figure 4 antibiotics-13-00252-f004:**
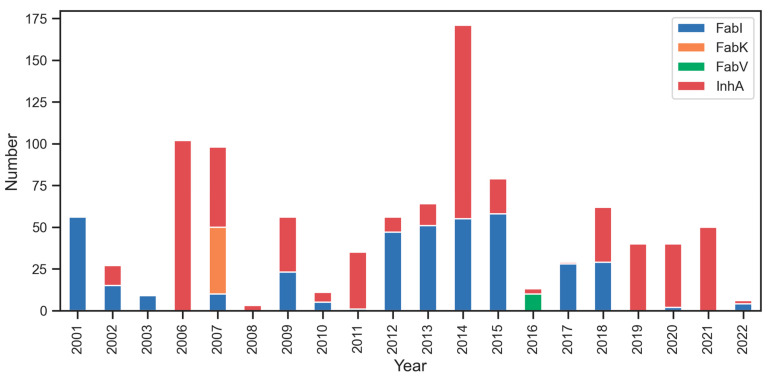
Temporal distribution of published research on Fab enzyme Inhibition. Notable research interest in InhA (red color) and FabI (blue color), contrasted with limited interest in FabK (orange color) and FabV (green color).

**Figure 5 antibiotics-13-00252-f005:**
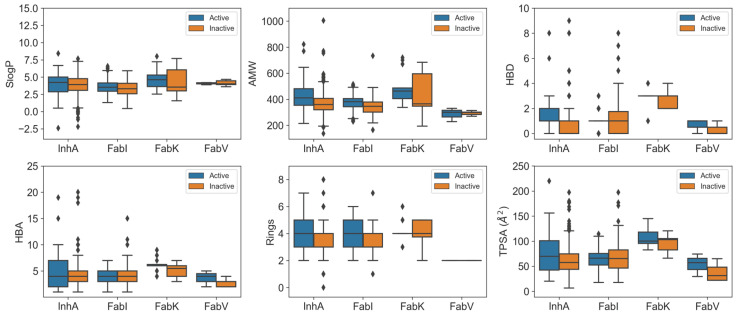
Boxplots comparing physicochemical characteristics of active (blue) and inactive (orange) inhibitors across InhA, FabI, FabV, and FabK enzyme datasets.

**Figure 6 antibiotics-13-00252-f006:**
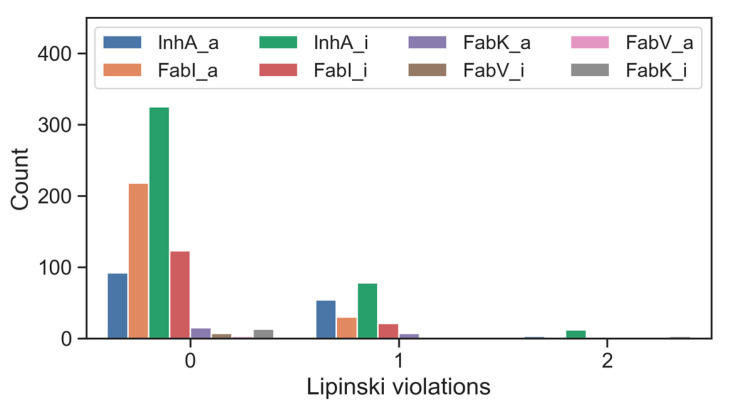
Number of inhibitors of individual enzymes, organized by activity and Lipinski violations; _a for active and _i for inactive.

**Figure 7 antibiotics-13-00252-f007:**
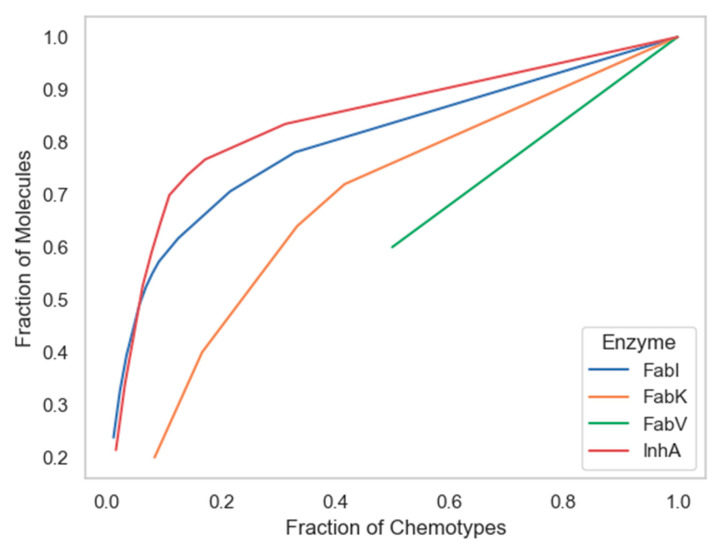
Cyclic System Retrieval (CSR) curves for four enzyme datasets.

**Figure 8 antibiotics-13-00252-f008:**
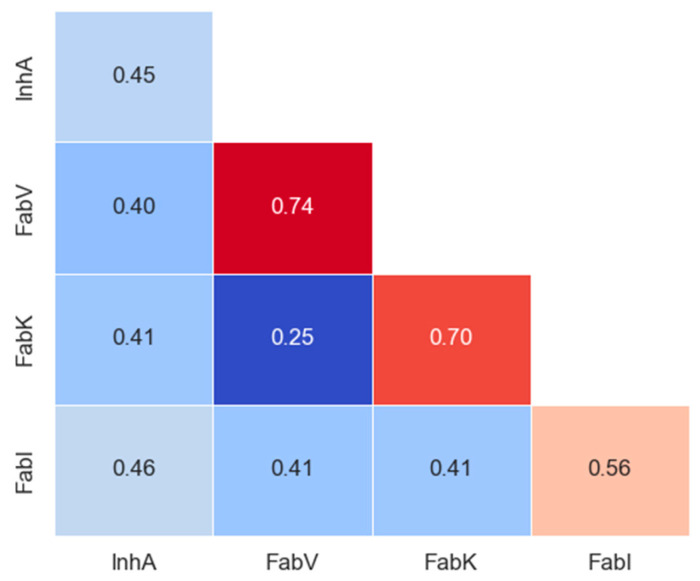
Intra- and inter-group Tanimoto similarity based on MACCS keys (166-bit) fingerprints. The diagonal in the matrix depicts intra-library comparisons, i.e., the similarity between inhibitors of enzymes. Red colors indicate larger similarity, while blue colors indicate smaller similarity.

**Figure 9 antibiotics-13-00252-f009:**
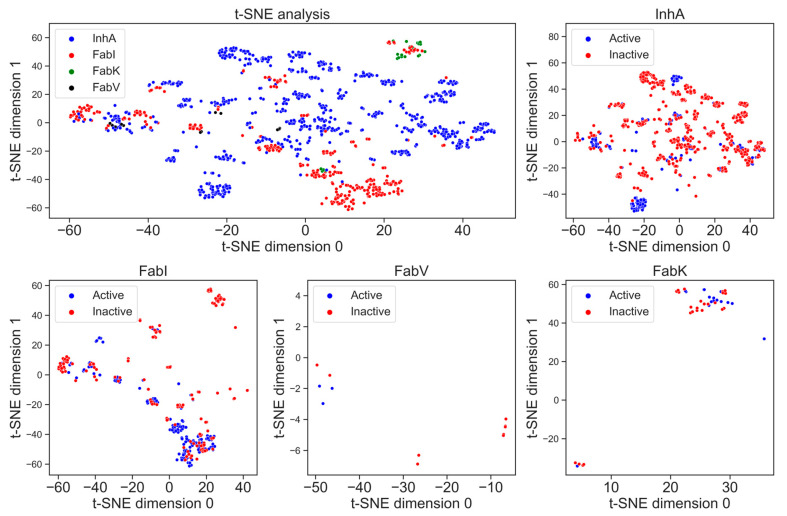
t-SNE analysis of the MACCS fingerprints for active (blue) and inactive (red) inhibitors of the enzymes InhA, FabI, FabV, and FabK.

**Figure 10 antibiotics-13-00252-f010:**
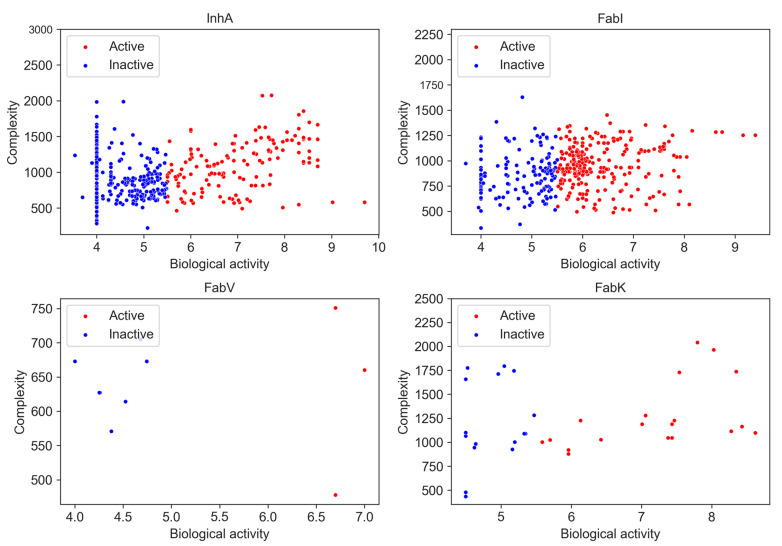
Relationship between BertzCT complexity index and pIC50 activity. Higher pIC50 values correspond to greater complexity.

**Figure 11 antibiotics-13-00252-f011:**
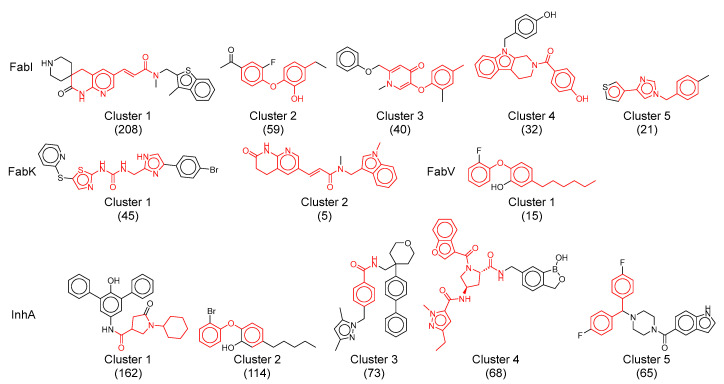
Representative structures of the largest clusters in each enzyme dataset. Maximum common substructure of compounds within each cluster is colored red. The numbers in brackets represent the number of compounds in each cluster.

**Figure 12 antibiotics-13-00252-f012:**
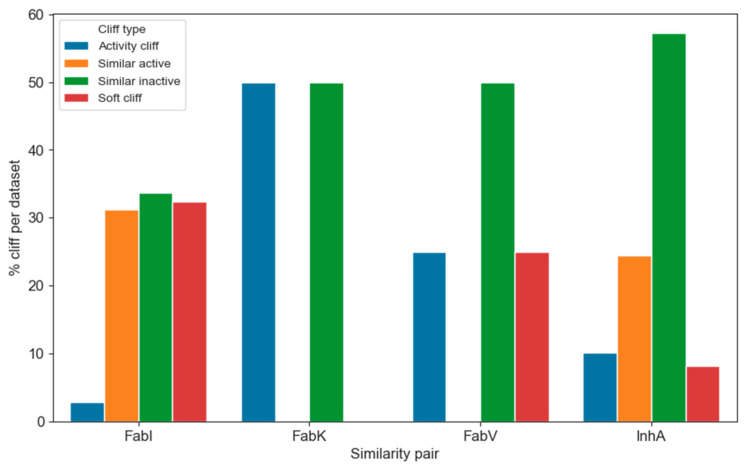
Distribution of the different types of activity cliffs in each enzyme dataset.

**Figure 13 antibiotics-13-00252-f013:**
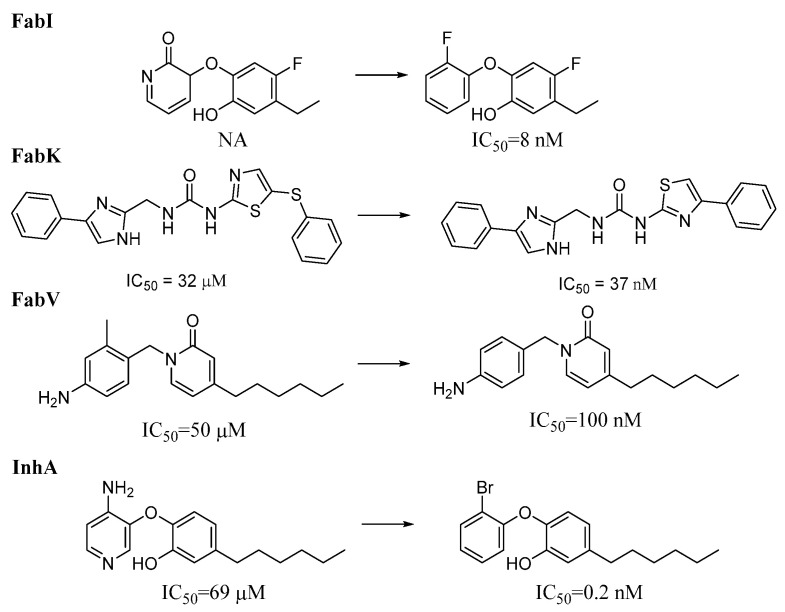
Representative activity cliffs within each enzyme dataset. NA = Not active, IC_50_ is the concentration of a compound, required to inhibit an enzyme’s activity by 50%.

**Figure 14 antibiotics-13-00252-f014:**
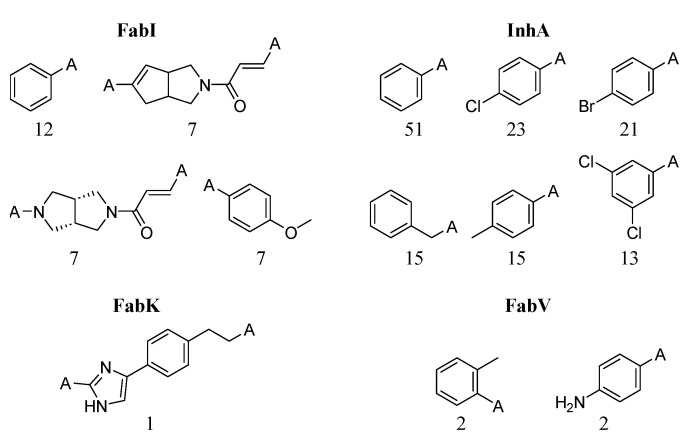
Most frequent fragments in transformation pairs. The numbers below each structure indicate the total number of fragments.

**Figure 15 antibiotics-13-00252-f015:**
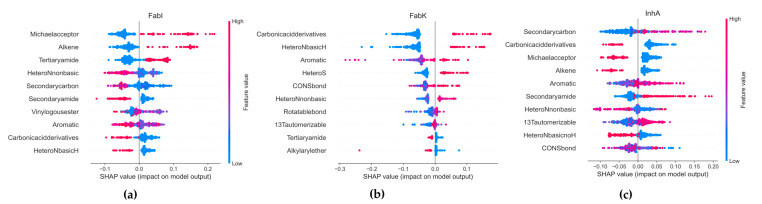
Results of the SMOTE analysis. (**a**) FabI inhibitors, (**b**) FabK inhibitors, (**c**) InhA inhibitors.

**Table 1 antibiotics-13-00252-t001:** Statistics on compounds categorized as PAINS and Brenk in all four enzyme datasets.

		PAINS		Brenk	
Enzyme	Activity	No.	Percent [%]	No.	Percent [%]
FabI	Active	30	7.1	175	12.4
	Inactive	18	1.3	76	5.4
FabK	Active	0	0	2	0.1
	Inactive	0	10.0	1	0.1
FabV	Active	1	0.1	2	0.1
	Inactive	1	0.1	7	0.5
InhA	Active	6	0.4	78	5.5
	Inactive	69	4.9	307	21.7

**Table 2 antibiotics-13-00252-t002:** Diversity analysis results of scaffolds in the four enzyme datasets: Scaled Shannon entropy (SSE) for the top 20 chemotypes and fraction of compounds within the most populated chemotypes.

Enzyme	N	M	N/M	N_sing_	N_sing_/N	N_sing_/M	AUC	F_50_
InhA	38	266	0.143	20	0.526	0.075	0.846	0.085
FabV	2	5	0.400	NA	NA	NA	0.4	NA
FabK	12	25	0.480	7	0.583	0.28	0.670	0.236
FabI	28	269	0.104	14	0.500	0.052	0.857	0.060

N, number of chemotypes; M, number of active molecules; N_sing_, number of singletons; AUC, area under the curve; F_50_, fraction of chemotypes that contains 50% of the dataset.

**Table 3 antibiotics-13-00252-t003:** Statistical analysis of the five largest clusters for InhA, FabI, FabV, and FabK.

Enzyme	Cluster	IntSim ^a^	%Cmp ^b^	nCl ^c^	nCl/nCmp ^d^	nCmp ^e^	%Actives	nCmpE ^f^	Q3 ^g^	pIC_50_ ^h^
FabI	1	0.30	46.7	22		208	77.4	445	6.6	9.4
FabI	2	0.43	13.3			59	72.9		6.9	8.1
FabI	3	0.41	9.0		0.049	40	47.5		5.8	6.9
FabI	4	0.11	7.2			32	75		6.7	6.7
FabI	5	0.29	4.7			21	33.3		5.9	6.6
FabK	1	0.30	88.2	3		45	53.3	51	7.6	8.6
FabK	2	0.08	9.8		0.059	5	20		5.1	5.3
FabV	1	0.40	100	1	0.067	15	33.3	15	6.4	7.0
InhA	1	0.28	16.4	77		162	38.9	990	6.2	7.2
InhA	2	0.52	11.5			114	36.0		7.0	9.7
InhA	3	0.48	7.4		0.078	73	45.2		6.5	7.7
InhA	4	0,20	6.9			68	95.6		8.4	8.7
InhA	5	0.47	6.6			65	10.8		5.1	6.0

^a^ IntSim—Intracluster similarity, ^b^ %Cmp—Proportion of compounds in an enzyme dataset, ^c^ nCl—Number of clusters, ^d^ nCl/nCmp—Number of clusters divided by the number of compounds, ^e^ nCmp—Number of compounds in a cluster, ^f^ nCmpE—Number of compounds in an enzyme dataset, ^g^ Q3—Third quartile of pIC_50_ values in a cluster, ^h^ pIC_50_—highest pIC_50_ value in a dataset.

## Data Availability

Data are contained within the article and Appendix A.

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
