# Peer review of "Navigating the Chemical Space of ENR Inhibitors: A Comprehensive Analysis"

_antibiotics, 2024, doi:10.3390/antibiotics13030252_

Round 1
Reviewer 1 Report
Comments and Suggestions for Authors
In this manuscript the authors describe a comprhensive computational investigation of compounds targeting ENR enzymes. This aproach is outstanding to clarify the essential feature for the design of active compounds.
The proposed methodology is of great interest, the only major weakness is the lacking of predictive value. I suggest to apply the selected feature to a small set of compounds designed on the basis of the obtained results in order to verify the predictive value of the computational investigation.
As minor issues:
The temporal analysis of figure 5 should be moved to introduction.
The N-methylpyridone structure of cluster 3 in figure 12 must be corrected.
The IC50 threshold of 5.5, please justify this choice, also with literature if available.
Author Response
We would like to thank the reviewer for his/her insightful comments on the manuscript, which have significantly contributed to enhancing the overall quality of our work's presentation. Below, I provide a response to each of the issues raised by the reviewer.
- The temporal analysis of figure 5 should be moved to introduction.
The temporal analysis presented in Figure 5 has been relocated to the introduction alongside the figure itself. Due to slight discrepancies in fitting the text seamlessly into the introduction's narrative, minor adjustments were made to this paragraph.
- The N-methylpyridone structure of cluster 3 in figure 12 must be corrected.
The structure was corrected.
- The IC50 threshold of 5.5, please justify this choice, also with literature if available.
Decision to set treshold of 5.5 was justified. The beggining of the Results section is now: »To distinguish between active and inactive compounds, we used a threshold of 5.5 for the pIC50 value. This decision was made in view of the fact that many articles in this field have chosen threshold values between 5 and 6. Furthermore, the choice of 5.5 as a threshold resulted in a distribution of 855 (61%) inactive and 557 (39%) active compounds in our database, ensuring a balanced representation of both categories [13,15,16].«
- The proposed methodology is of great interest, the only major weakness is the lacking of predictive value. I suggest to apply the selected feature to a small set of compounds designed on the basis of the obtained results in order to verify the predictive value of the computational investigation.
I would like to ask the reviewer to clarify this comment further because I believe I do not fully comprehend it. While our RF model was not originally intended for predictive purposes, we could easily design a small dataset and utilize RF to predict the activity of each compound within it. However, to ensure reliability, this small dataset would need to be within the domain of applicability, meaning it should be similar to the training set used. This would result in two classes of compounds: active and inactive. Nevertheless, I am uncertain about the subsequent steps to take.
When comparing the physicochemical properties of active compounds to those in the original training dataset, I encounter at least two challenges. Firstly, although both datasets may contain similar compounds, the distribution of their physicochemical properties could differ. Secondly, our RF model was trained on a substructure count fingerprint, which does not encompass all the physicochemical properties we examined in our study. Therefore, comparing the physicochemical properties of active compounds between datasets may yield similarities, but I am unsure if this is sufficient to verify predictive value.
Utilizing this RF model to predict the activity of compounds and integrating these results into another MMP could provide significant utility, as it would enhance the effectiveness of MMP analysis in molecular optimization navigation. Indeed, a similar approach has been recently described (Yang et al., Journal of Cheminformatics 2021, 13(1): https://doi.org/10.1186/s13321-021-00564-6), but I believe it falls beyond the scope of our research.
I would also like to address the complaint of another reviewer that commented the density of figures and information in our manuscript and supplementary material. While we have made efforts to shorten the article due to its length, I acknowledge that additional analysis could be beneficial. However, it is essential that any supplementary analysis is concise.
Reviewer 2 Report
Comments and Suggestions for Authors
Kuralt & Frlan present a comprehensive and exhaustive chemography of ENRs. The work has been throughly conducted. However I find that both the manuscript and supplementary material are quite heavy on figures and information.
A careful assessment of data should be made, as in its present form the manuscript could prove a difficult read to the uninitiated audience.
Also, several tests or methods are not introduced properly. Again, to an audience with a strong background in cheminformatics some of the choices can be standard protocol, still some guidelines should be provided.
Similarly, some sections can use supporting references, for instance, when presenting the rule of five the authors correctly state their use and limitations. But it seems rather anecdotal without a citation. There are notable examples where the rule of five has been challenged and/or clearly debunked, consider the addition of at least one.
A major suggestion I have is the inclusion of bullet points summarizing the highlights of the chemography.
Comments on the Quality of English LanguageThere are some grammatical errors in the manuscript.
Author Response
We would like to thank the reviewer for his/her insightful comments on the manuscript, which have significantly contributed to enhancing the overall quality of our work's presentation. Below, I provide a response to each of the issues raised by the reviewer.
- Kuralt & Frlan present a comprehensive and exhaustive chemography of ENRs. The work has been thoroughly conducted. However I find that both the manuscript and supplementary material are quite heavy on figures and information.
We have addressed the concern regarding the heaviness of figures and information in both the manuscript and supplementary material. Specifically, we have condensed the molecular similarity paragraph by shortening the discussion on ECFP4 fingerprints and moving Table 3 to the supplementary information. Additionally, Figure 5 has been relocated to the supplementary information as Figure S1. Similarly, the chemotype diversity paragraph has been made shorter, and discussions on topics like Shannon entropy and the fraction sp3 have been removed from the manuscript as they were deemed less relevant to our analysis."
- A careful assessment of data should be made, as in its present form, the manuscript could prove a difficult read to the uninitiated audience
- We appreciate the reviewer's concern about the readability of our manuscript, especially for readers who may not be familiar with the topic. To address this, we have restructured our presentation to enhance clarity and accessibility. Specifically, we've made revisions to the SHAP chapter, relocating machine learning results to the Methods section and seamlessly integrating them with the existing SHAP analysis description. This restructuring not only streamlines the article but also ensures that all readers, regardless of their background, can easily follow our analysis.
- Furthermore, we've taken steps to provide comprehensive explanations for the tests and methods used, making sure that individuals with varying levels of expertise in cheminformatics can understand the content. In line with this, we've made adjustments to various sections of the manuscript:
- The molecular similarity paragraph has been shortened, focusing solely on MACCS fingerprints for our analysis. Discussion on ECFP4 fingerprints has been moved to the supplementary information, along with Table 3, which contains detailed statistics. Only the similarity matrix remains in this chapter, enhancing its clarity.
- Figure 5 has been relocated to the supplementary information as Figure S1.
- The chemotype diversity paragraph has also been condensed, with the removal of a discussion on Shannon entropy, which was deemed less critical for our analysis.
- The complexity of datasets, including the fraction sp3, which wasn't suitable for our analysis, has been omitted from the manuscript.
- We've emphasized that the Lipinski rule of five serves as a guideline rather than a strict rule, discouraging overly dogmatic adherence to it.
- The chapter on MMP has been shortened, removing a paragraph on the importance of statistical evaluation, as it was considered self-evident. Discussions on the number of cuts and a figure describing these cuts in each dataset have been moved to the supplementary information, along with a table detailing the most common MMP transformations.
- These adjustments aim to enhance the overall readability and accessibility of our manuscript while maintaining the integrity and depth of our analysis.
- Also, several tests or methods are not introduced properly. Again, to an audience with a strong background in cheminformatics some of the choices can be standard protocol, still some guidelines should be provided.
We agree with the reviewer. In the Results section, the SHAP chapter has been revised. The results of the machine learning protocols have been relocated to the Methods section and integrated with the existing description of the SHAP analysis. It was determined that the machine learning results, by themselves, are not essential for comprehending the analysis. This adjustment has resulted in a shorter article, allowing readers with a strong background to refer to the Methods section, where the preparation of the dataset and the procedure for feature selection are described. Additionally, the MMP chapter has been condensed, and details regarding the statistical analysis of the MMP protocol results, such as the number of MMPs obtained, have been moved to the Supplementary Information (SI) for brevity.
- Similarly, some sections can use supporting references, for instance, when presenting the rule of five the authors correctly state their use and limitations. But it seems rather anecdotal without a citation. There are notable examples where the rule of five has been challenged and/or clearly debunked, consider the addition of at least one.
We appreciate the reviewer's suggestion to include supporting references in sections where appropriate. To strengthen the discussion on the Lipinski rule of five, we have incorporated additional references, including original publications by Lipinski that elucidate the principles of the rule. Moreover, we have expanded on the explanation of the rule's limitations, emphasizing its role as a guideline rather than an absolute rule. This enhancement aims to provide readers with a more comprehensive understanding of the topic.
- A major suggestion I have is the inclusion of bullet points summarizing the highlights of the chemography.
Regarding the suggestion for including bullet points summarizing the highlights of the chemography, we have opted to modify the introduction paragraph to emphasize the importance of different techniques used in navigating chemical space of ENR inhibitors. While we appreciate the suggestion, we believe that bullet points might shift too much attention to chemometry alone, potentially overshadowing other essential techniques like bioinformatics and cheminformatics. Given that our work primarily focuses on chemometrics, we have chosen to dedicate two paragraphs in the introduction to describing the methods relevant to our research. This approach provides readers with a clear overview of the interdisciplinary methods employed in our study without delving into exhaustive detail on each technique.
We believe that these revisions enhance the manuscript's scientific rigor and relevance, bridging computational analysis with experimental data to offer a comprehensive understanding of ligand-target interactions.
Reviewer 3 Report
Comments and Suggestions for Authors
The paper entitled "Navigating the Chemical Space of ENR Inhibitors: A Comprehensive Analysis" reports a cheminformatic analysis of ENR inhibitors. The analysis is well-performed and the obtained outcomes can be exploited for the design of new inhibitors. I suggest to publish this manuscript after that the following point has been addressed.
- The SAR discussion must be enriched by including a description on how the findings of this analysis support the available experimental data concerning the interaction between ligand and target.
Author Response
We would like to thank the reviewer for his/her insightful comments on the manuscript, which have significantly contributed to enhancing the overall quality of our work's presentation. Below, I provide a response to each of the issues raised by the reviewer.
- The SAR discussion must be enriched by including a description on how the findings of this analysis support the available experimental data concerning the interaction between ligand and target.
In response, we have incorporated a concise discussion in the final chapter, emphasizing the correlation between our computational analysis outcomes and experimental observations. Specifically, our analyses revealed common transformations in ligand-target interactions, underscoring the importance of specific functional groups, particularly in the InhA and FabI datasets. These findings, supported by crystal structures and previous experimental studies, provide valuable insights into structure-activity relationships and inform rational drug design strategies effectively.
We believe that these revisions enhance the manuscript's scientific rigor and relevance, bridging computational analysis with experimental data to offer a comprehensive understanding of ligand-target interactions.
Round 2
Reviewer 1 Report
Comments and Suggestions for Authors
The authors addressed all requested issues.
Concerning the request of of clarify the comment, I'd like to see exactly wath the authors proposed "Design a small dataset and utilize RF to predict the activity of each compound within it."
I agree that this could be out the scope of this work. I strongly encourage the authors to implement a system with general predictive value in the next future.